# A temporary indwelling intravascular aphaeretic system for in vivo enrichment of circulating tumor cells

Tae Hyun Kim [1,2,3,4], Yang Wang[1,3], C. Ryan Oliver[4], Douglas H. Thamm[5], Laura Cooling[6], Costanza Paoletti[7], Kaylee J. Smith[1], Sunitha Nagrath [1,3] & Daniel F. Hayes [7]

Circulating tumor cells (CTCs) have become an established biomarker for prognosis in patients with various carcinomas. However, current ex vivo CTC isolation technologies rely on small blood volumes from a single venipuncture limiting the number of captured CTCs. This produces statistical variability and inaccurate reflection of tumor cell heterogeneity. Here, we describe an in vivo indwelling intravascular aphaeretic CTC isolation system to continuously collect CTCs directly from a peripheral vein. The system returns the remaining blood products after CTC enrichment, permitting interrogation of larger blood volumes than classic phlebotomy specimens over a prolonged period of time. The system is validated in canine models showing capability to screen 1–2% of the entire blood over 2 h. Our result shows substantial increase in CTC capture, compared with serial blood draws. This technology could potentially be used to analyze large number of CTCs to facilitate translation of analytical information into future clinical decisions.

[1] Department of Chemical Engineering, University of Michigan, Ann Arbor, MI 48109, USA. [2] Department of Electrical Engineering and Computer Science, University of Michigan, Ann Arbor, MI 48109, USA. [3] Biointerfaces Institute, University of Michigan, Ann Arbor, MI 48109, USA. [4] Biomedical Engineering, University of Michigan, Ann Arbor, MI 48109, USA. [5] Flint Animal Cancer Center, College of Veterinary Medicine and Biomedical Sciences, Colorado State University, Fort Collins, CO 80523, USA. [6] Department of Pathology, University of Michigan, Ann Arbor, MI 48109, USA. [7] Department of Internal Medicine, University of Michigan Rogel Cancer Center, Ann Arbor, MI 48109, USA. Correspondence and requests for materials should be addressed to S.N. (email: snagrath@umich.edu) or to D.F.H. (email: hayesdf@umich.edu)

Cancer metastases arise from circulating tumor cells (CTCs) that are shed from the primary tumor and circulate through lymphatic channels and blood[1]. Although identified more than 150 years ago[2], until recently, CTCs were difficult to detect, enumerate, and characterize. Using modern technologies, several studies have now demonstrated that elevated levels of CTC isolated from a single blood draw are prognostic in patients with metastatic breast, colorectal, prostate, and lung cancers, as well as early stage breast and prostate cancers[3–8]. Furthermore, CTC analysis holds promise for predicting benefit from targeted therapies, pharmacodynamic monitoring during treatment, and insight into the biology of metastases[9,10]. Indeed, CTC evaluation might be used for early detection of malignancy, if an assay with sufficient sensitivity and specificity could be developed.

CTCs are extremely rare events. For example, in a single 7.5 mL tube of whole blood drawn from an average patient with metastatic breast cancer, it is unusual to identify more than 10 CTCs within the context of billions of erythrocytes and millions of leukocytes normally present. More than a hundred ex vivo CTC capture devices have been developed to enrich and isolate CTC from whole blood[10–12]. However, CTC isolation using these technologies is limited to small blood volumes (usually 1–50 mL) due to patient safety concerns, and therefore the absolute number of CTC is small. Moreover, a single blood draw interrogates only those CTC present at the time of venipuncture, and does not take into account temporal differences in CTC release into the circulation. The ability to interrogate larger blood volumes over extended periods of time might enhance the number of CTC available for enumeration, which would therefore increase statistical confidence of sampling for comparison of serial levels[13,14]. This approach would also provide more CTCs for molecular phenotyping, genotyping, and further biological characterization.

Attempts to increase the volume of blood evaluated for CTC isolation have included using alternative sites of blood collection, including the vessels draining primary cancers accessed at the time of surgery[15]. However, the accessibility to these sources is limited according to the location of the tumor, and this approach is not practical for routine diagnostic use. Furthermore, despite the considerable number of CTCs detected in samples from the tumor draining vessels, many cells that are disrupted during surgery rapidly undergo apoptosis[16], and their biological and clinical impact is unknown.

Other investigators have reported isolating CTC in cytopheresis products, either from whole blood or bone marrow, often collected in anticipation of hematopoietic stem cell transplantation therapy[17–19]. Although this strategy enables a substantial increase in detecting CTCs compared with a single blood draw, standard cytopheresis is cumbersome and inconvenient for the patient. Furthermore, cytopheresis products mainly consist of concentrated peripheral blood mononuclear cells, which require an additional high throughput screening step for CTC identification. As with cannulating tumor-draining vasculature, the logistics required for standard leukapheresis/cytopheresis render this approach impractical as a standard diagnostic test, especially for application in a serial fashion.

Investigators have also reported use of an intravenous gold-coated stainless steel medical wire with a hydrogel layer covalently coupled with antibodies against epithelial cellular adhesion molecule (EpCAM) protein (GILUPI CellCollector)[20–25]. However, physiologic variations between patients affecting blood flow and affinity make it difficult to standardize quantitative interpretation of CTCs by time of insertion. Similarly, a recent study has demonstrated in vivo capture of non-small cell lung cancer cells injected into a porcine model, using a flexible magnetic wire (MagWIRE)[26]. However, the approach requires pre-injection of EpCAM coated magnetic particles to label CTCs which limits its long-term application due to possible systemic exposure of iron overload.

To address the shortcomings of current approaches, we developed a temporary indwelling, intravascular aphaeretic CTC isolation system that can potentially be worn by a patient for several hours and through which a relatively large volume of blood can be interrogated. We demonstrate the feasibility of this device in ex vivo experiments, and we have performed proof-of-principle investigations of its potential clinical application for in vivo CTC capture in a canine model.

## Results

**Portable aphaeretic system design.** The overall design of the system is illustrated in Fig. 1a. Each functional component is integrated into a compact 3D printed manifold (Fig. 1b) to permit portability and is controlled through a custom built mobile application via wireless communication. The system consists of four major parts: a micro-controller, peristaltic pump, heparin

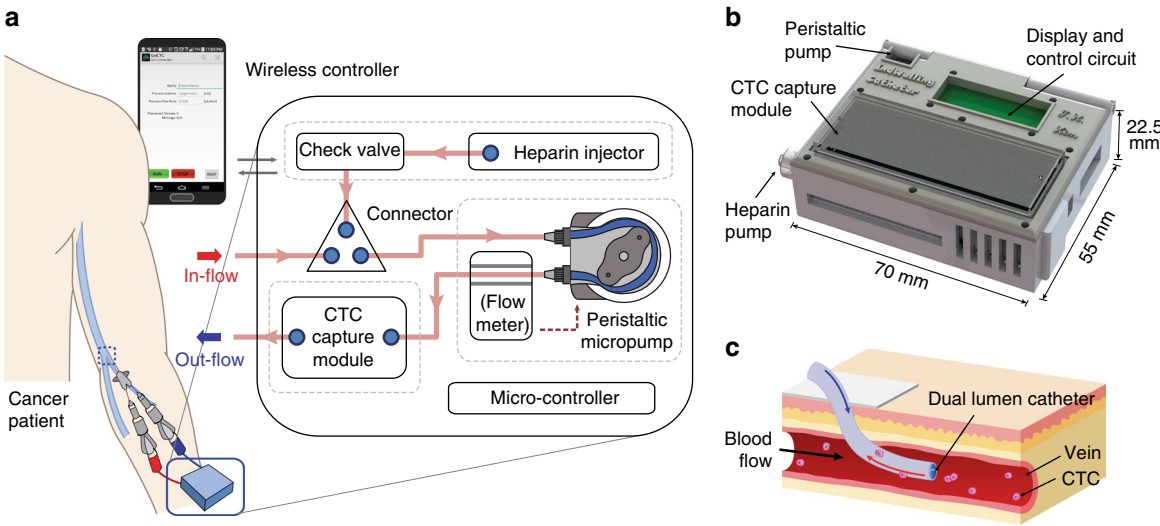

**Fig. 1** The in vivo aphaeretic CTC isolation system. **a**, **b** Schematic overview of the indwelling intravenous system by functional components (**a**) and manifold (**b**). **c** Application of a dual lumen catheter for in vivo CTC isolation

injector, and a CTC capture module that contains a microfluidic CTC capture chip. The system is designed to accommodate any type of CTC isolation device, as long as it is configured to fit into the system manifold.

Whole blood is routed into the system from a subject's peripheral vein with a single cannulation using a dual-lumen catheter (Fig. 1c) via the efflux lumen (designated as in-flow in Fig. 1a) then through the CTC capture module, exiting back into the subject's circulatory system via the influx lumen of the catheter (labeled as out-flow in Fig. 1a). Each end of the catheter is connected to a silicone tube, treated with anticoagulation reagents, with luer lock adaptors that thread into a peristaltic pump and CTC capture module forming a closed loop structure. The blood flow is driven by the peristaltic pump with a pre-programmed flow rate and total processing volume. In between, a flow rate sensor is implemented to monitor and maintain a constant flow through a feedback loop. To prevent blood clot formation during operation, heparin is continuously infused through an injection pump. Every unit in direct contact to the blood during sampling and re-transfusion, is sterilized and individually inspected before use and disposed afterwards.

**$^{HB}$GO chip design and function**. The microfluidic herringbone graphene oxide CTC chip ($^{HB}$GO chip) is designed using functional graphene oxide sheets for sensitive capture and chaotic mixing via herringbone structures for enhanced throughput. It consists of a $24.5 \times 60$ mm silicon dioxide substrate with patterned gold thin film layer bonded to a polydimethylsiloxane (PDMS) structure containing four bifurcating microchannels with herringbone grooves embedded on their top surface (Fig. 2a). Functional graphene oxide nano sheets are assembled onto the

gold layer presenting high-density anti-EpCAM antibodies on the substrate surface through chemical cross-linkers[27,28]. The geometry of the herringbone structure has been determined based upon previous designs used for chaotic mixing at low $Re$ number[29,30]. However, unlike earlier devices in which the interaction mainly takes place near the grooves[31], the geometry of the herringbone structure has been modified to maximize the contact frequency of cells to the substrate where the antibodies are tethered (Fig. 2b).

Twenty-four chevrons, a set of twelve staggered asymmetrically, was defined as a single mixing unit and periodically shifted along the channel axis to place each vertex points with a spacing of 25 μm. The distribution of these points, where a vertical drag force is induced by adjacent micro vortexes, increased the probability of cells to be directed toward the antibody coated substrate. The dimension of the groove height, width, and pitch was also adjusted to decrease the hydraulic resistance past that of the main fluidic channel. This unbalanced resistance between the channel and grooves increased the overall fluidic circulation by deflecting a significant portion of fluid and cells into the herringbone structure (Supplementary Fig. 1). Cells immersed and guided through the herringbone moved in a zigzag trajectory until captured, which increased its traveling time and distance within the chip (Supplementary Fig. 2). The final dimension of the PDMS structure was as followed: overall height of the main fluidic channel was 40 μm, with a groove height set to 60 μm; the groove pitch and width was 200 and 160 μm, respectively, and the angle between the chevrons was 45° (Fig. 2b).

**Evaluation of $^{HB}$GO chip using cancer cell lines**. To validate the performance of the $^{HB}$GO chip for CTC capture at high flow

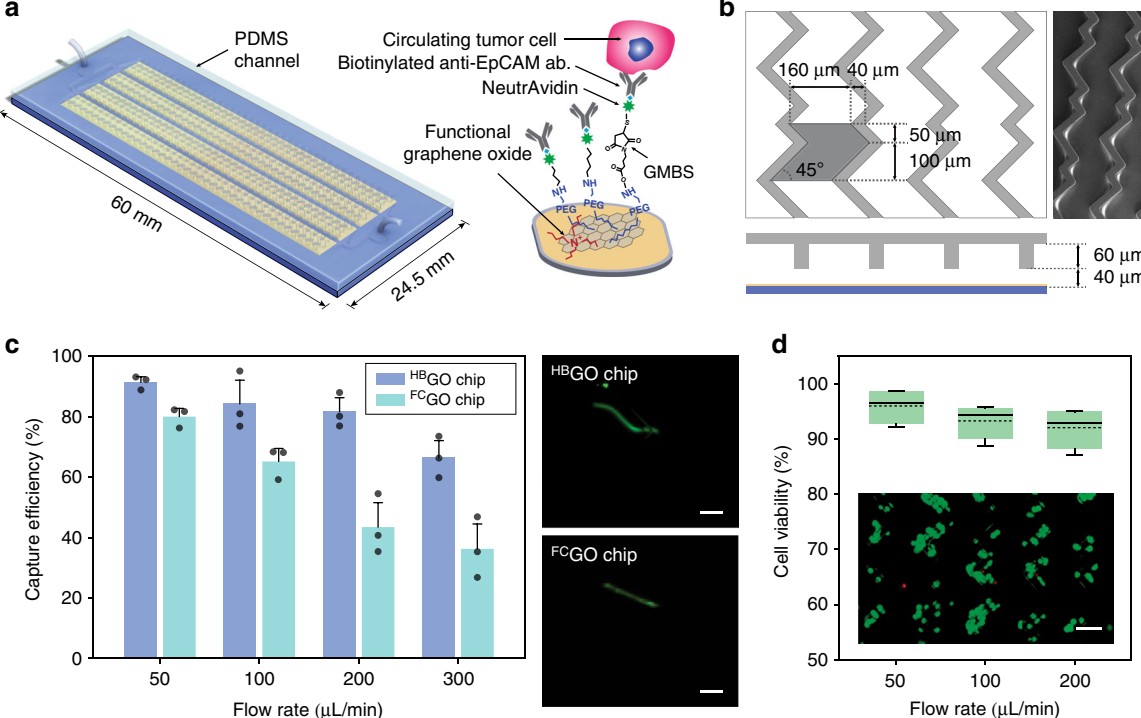

**Fig. 2** Ex vivo evaluation of $^{HB}$GO chip for CTC capture. **a** Schematic of the $^{HB}$GO chip and the conjugation chemistry between functional graphene oxide and anti-EpCAM antibody. **b** Schematic and micrograph of the herringbone grooved channel geometry. **c** CTC isolation performance of the $^{HB}$GO chip compared with the $^{FC}$GO chip (data points are means ± s.d., $n = 3$). A fluorescent labeled single MCF7 cell traveling within the two different channels shows difference in cell trajectory. **d** Measure of cell viability as a function of flow rate after capture (dotted line is the mean, $n = 4$). Inset: fluorescence microscope image of MCF7 cells stained with live/dead assay. Viable cells are shown in green. All scale bars represent 100 μm. Source data are available in the Source Data file

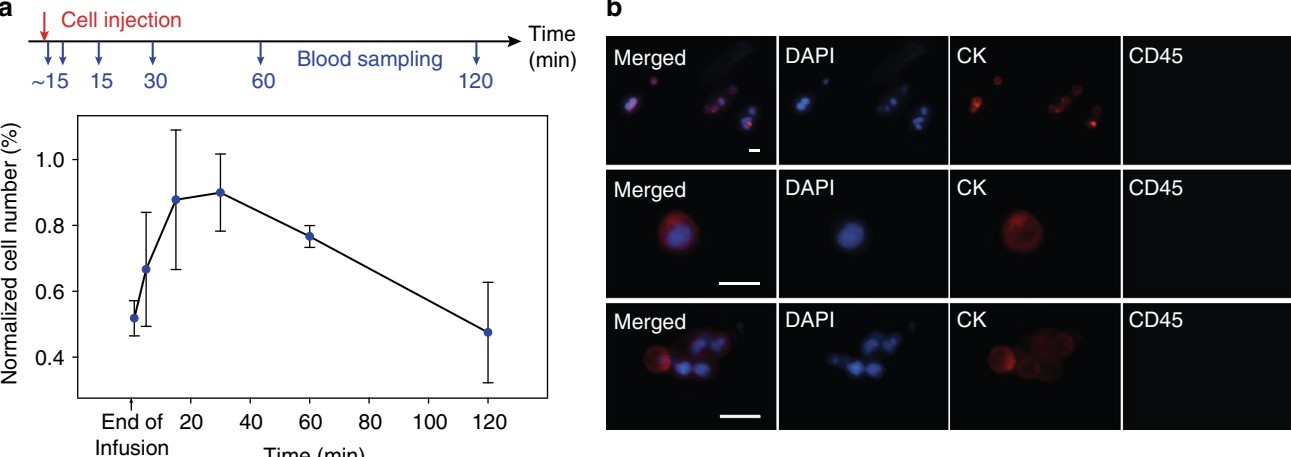

**Fig. 3** Cellular kinetics of MCF7 cells after intravenous injection in dogs. **a** Overview of cell injection followed by venipuncture for blood sampling and ex vivo enumeration with the $^{HB}$GO chip. Normalized number of MCF7 cells isolated from 1 mL serial blood samples, as a function of time after infusion, were averaged from three separate experiments (data points are means ± s.d., $n = 3$). **b** Fluorescent microscope image of MCF7 cells and clusters captured ex vivo. Cells were stained with DAPI (blue), Cytokeratin (red), and CD45 (green). The scale bars represent 25 μm. Source data are available in the Source Data file

rates, a flat channel graphene oxide chip ($^{FC}$GO chip)[27,28] previously reported was used for comparison (Fig. 2c, Supplementary Fig. 3). Cultured human breast cancer MCF7 cells were labeled with a fluorescent cell-tracker dye and spiked into 5 mL of PBS buffer solution with a concentration of 50–200 cells/mL. Both cells captured on-chip and non-captured cells collected into a waste well were counted after processing the samples to calculate the capture efficiency based on the total number of cells. At a flow rate of 1 mL/h (≈16.67 μL/min), a range in which most affinity-based microfluidic devices operate, both chips showed high capture efficiency with a mean yield above 90%. However, with increasing flow rates, the average cell capture efficiency for the $^{FC}$GO chip dramatically dropped below 80% at flow rates of 50 and 100 μL/min, and further decreased below 50% at 200 μL/min and higher. In contrast, the $^{HB}$GO chip maintained a target yield of >80% up to 200 μL/min with no significant decrease in overall capture efficiency, indicating the effect of the herringbone mixer for improved cell surface interaction.

In addition, cell viability was assessed at different flow rates to determine the effect of shear force induced by increasing flow rates during the isolation process (Fig. 2d). This was one of the critical readouts as the viability could adversely affect further downstream analysis of the isolated cells. Up to a flow rate of 200 μL/min, >90% of cells were found to be viable with no significant reduction compared with the viability at lower flow rates. The cell viability significantly decreased (<70%) when higher flow rates were applied, therefore becoming the rate limiting factor of the $^{HB}$GO chip.

Finally, to examine the effect of UV on chemical cross linkers and antibodies during the final sterilization process, chips were exposed to varying doses of UV and subsequently tested for cell immobilization (Supplementary Fig. 4). Compared to no exposure, no adverse effect was observed in the cell capture performance. In addition, the endotoxin levels of fluids obtained from the chips after sterilization were less than the 0.5 EU/mL detection limit, with no bacterial growth observed after plated and cultured on sheep blood agar.

**Ex vivo capture of intravenously infused CTCs in canine.** To translate the concept of in vivo CTC capture into human clinical trials using the system, the feasibility of this approach was validated in canine models. Dogs were chosen over murine models, since their relatively larger vascular size and blood volume allowed easy and reliable venous access, and dogs may represent a much more faithful model of human cancer[32,33]. A total of $2 \times 10^7$ non-labeled MCF7 cells were injected intravenously to mimic the occurrence of natural CTCs in blood. To estimate the cell distribution during circulation and determine the optimal time interval for isolation, peripheral blood collected by venipuncture was sampled before and after 1 (end of infusion), 5, 15, 30, 60, and 120 min following injection. The sampled whole blood was then processed through the $^{HB}$GO chip ex vivo at a flow rate of 100 μL/min. Captured MCF7 cells were quantified by positive CK staining with the absence of canine CD45 (Supplementary Fig. 7). To take into account for false positive CK staining from canine leukocytes, counting results were subtracted from that of the blood sampled before cell infusion and normalized to the maximum cell counts from each test. Figure 3a illustrates cellular removal kinetics of MCF7 cells after intravenous infusion, averaged from three separate investigational animals on different days (Supplementary Fig. 8, Supplementary Table 1), as determined by serial blood draws. The distribution time for cells to appear in circulation after injection was less than a minute. MCF7 cell counts rose to a maximum at 30 min from inoculation and declined over the succeeding 90 min due to their clearance during circulation. Most MCF7 cells were identified as single cells but clusters were also detected (Fig. 3b). Importantly, although the expression level of CK decreased as the time of blood sampling increased, MCF7 cells were detectable throughout the duration of the pre-planned experimental duration of 2 h. No short-term or long-term adverse effects from the MCF7 injection and venipuncture were observed.

**In vivo capture of intravenously infused CTC.** Following establishment of appropriate time frames for both efficacy and safety and demonstration of cell capture ex vivo, the entire system was tested for direct cell harvesting from in vivo circulation. Here, the flow rate sensor was excluded from the system since it was not disposable after use. After placement of a double-lumen catheter into the jugular vein, the system was attached to the $^{HB}$GO chip and heparin pump via pre-sterilized extension sets. All fluid paths were primed with 1% heparin and connected to the catheter (Supplementary Fig. 10). Prior to cell injection, the system was

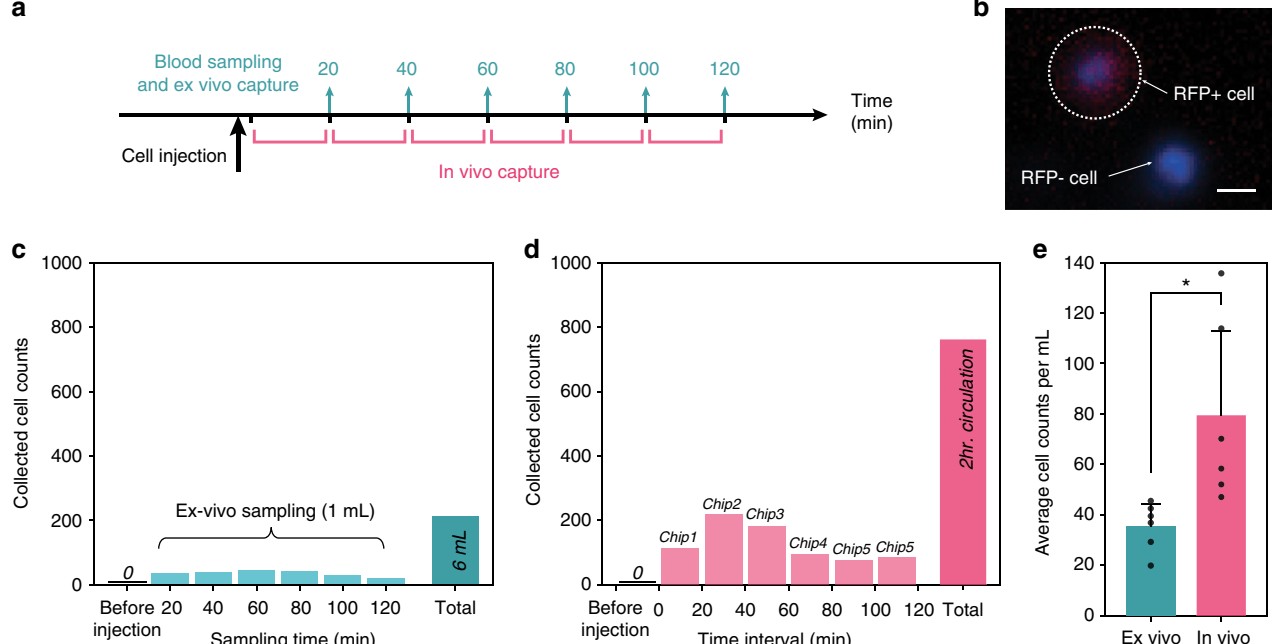

**Fig. 4** Comparison of MCF7 cells detected by ex vivo versus in vivo CTC isolation method. **a** Schematic illustrating the workflow of the experiment. **b** Fluorescence microscope image of RFP transfected MCF7 cell with RFP negative leukocyte. The scale bar represents 10 μm. **c** Number of MCF7 cells collected from six serial sampling of blood and its total by ex vivo isolation using the $^{HB}$GO chip. **d** MCF7 counts from six serial chips replaced during in vivo cell enrichment and its total number of cells. **e** Comparison between total number of cells recovered per mL from ex vivo versus in vivo isolation method (data points are means ± s.d., $n = 3$, two-tailed paired $t$-test $P < 0.05^*$, $t = 3.39$). Source data are available in the Source Data file

activated for 30 mins to inspect for any blood clotting or clogging activity in the device, tube, and connector junctions. By varying the heparin concentration during operation, a 10% concentration of heparin with a 1:5 volume ratio to the blood induced no detectable blood clogs or clots. The peristaltic motion of the pump, as well as the micro vortices generated within the chip enabled heparin to be gently mixed with blood efficiently.

Next, $2 \times 10^7$ red fluorescent protein (RFP) expressing MCF7 cells were injected into the cephalic vein of the canine subject, and the system was turned on after 1 min. Blood was then permitted to circulate through the device for up to 120 min. To compare and evaluate the system performance, 1 mL of blood was also concurrently drawn through the same jugular catheter every 20 min and analyzed for MCF7 presence by ex vivo CTC capture (Fig. 4a). A micrograph of a captured CTC and a contaminating leukocyte, demonstrating the clear distinction between the two, is illustrated in Fig. 4b. The average number of cells captured ex vivo for each draw of blood was 35.33 ± 8.46 cells/mL with a maximum concentration of 45 cells/mL after 60 min (Fig. 4c). In total, 212 MCF7 cells were isolated and enumerated in the 6 mL of whole blood collected in 1 mL increments over the 2 h experiment. The slightly lower concentration and shift of time at which maximum concentration occurred, when compared with the cell injection study described above, was most likely due to the different physiological kinetics (differences in body size (7 kg), cardiac output, volumes of distribution, etc.) among the tested subjects.

In contrast to the ex vivo CTC capture, a total of 762 MCF7 cells were isolated in vivo over 2 h within the indwelling intravascular CTC capture system (Fig. 4d). The total capture of CTC in vivo by the indwelling system was approximately 3.5 times compared with the periodic blood draw approach (Figs. 4c and 4d). Importantly, the recovery rate measured as total number of cells per mL from in vivo circulation was also substantially greater compared with the ex vivo capture experiment ($P = 0.02$, Fig. 4e).

This difference was attributed to the varying concentration of MCF7 cells undergoing natural elimination with the observed blood half-life of 2 h in the immunocompetent host. As such, more discrete blood sampling at later times after the predicted maximum cell concentration resulted in a lower recovery rate than the continuous intravenous isolation done in vivo. No RFP positive cells were identified in the blood obtained before cell infusion, confirming the specificity of our counts.

In case the surface of the $^{HB}$GO chip became saturated with CTCs when applied for an extended period of time, we designed the system to be capable of instantly replacing the CTC capture module by swapping the chips, while maintaining sterility. This strategy permitted longer interrogation of increased blood volume without having to discontinue the intravenous access. In this regard, no visible interruption in blood flow was observed after swapping the chip every 20 min. As expected from the cellular kinetic data (Fig. 3a), CTC harvest was greatest in the second chip, which ran between the 20 to 40 min time period (Fig. 4d). There were no changes in clinical observations (temperature, pulse, respiration, body weight, food intake) or clinical pathology measurements (complete blood count, chemistry profile, coagulation) in any of the dogs at any time during or up to 7 days following the procedure.

## Discussion

Despite the promising clinical implications of monitoring CTCs as a liquid biopsy, current CTC assays are hampered by the requirement that only a limited amount of blood can be collected at a single time-point. The indwelling intravascular aphaeretic CTC collection system we describe interrogates a larger blood volume in vivo over an extended period of time for capture and subsequent analysis of CTC. We liken this device to a Holter monitor commonly used by cardiologists to monitor cardiac arrhythmias over a prolonged period of time vs. at a single time-point provided by a rhythm strip.

The system incorporates a CTC chip module which takes advantage of the functional graphene oxide immobilized with anti-EpCAM antibody and herringbone micro-channels. This chip has substantially increased sensitivity and throughput compared with previous immuno-affinity based CTC isolation platforms[11].

We have validated our approach in a canine model by successfully interrogating 1–2% of the entire canine blood volume over 2 h to isolate cancer cells in vivo. Major concerns regarding sterility, intra-device clotting, as well as intravascular thrombosis have been resolved by several strategies, including rigorous sterilization procedures and incorporation of a continuous heparin infusion injector. Furthermore, device clogging anticipated by saturating the CTC capture module with CTCs has been solved by the design of interchangeable CTC chips. This feature suggests that the system could be left in place for longer periods of time, potentially permitting even higher blood volume interrogation at little or no harm and minimal patient inconvenience. Further, previous in vivo CTC isolation strategies using standard cytophoresis necessitate bedside peripheral blood collection and are encumbered with potential blood cell loss. Our system is wearable, thereby allowing full patient mobility during operation with minimal cell loss.

It is not clear whether CTC dissemination from existing tissue metastases occurs at a continuous rate or, alternatively, in periodic, discrete segments. Likewise, it is not known whether CTCs can gain access to circulation equally from all metastases or in a non-uniform site-specific distribution. Current methods of CTC enumeration require assumption of homogeneous CTC distribution at the time of venipuncture. Our system permits periodic CTC assessment by swapping the CTC-collection chips while at the same time continuously monitoring blood flow. Similar to our results, previous studies have shown that the CTC counts per mL from diagnostic leukapheresis products were significantly higher than their matched peripheral blood samples when both are processed using the CellSearch® system[17]. Of note, the time-dependent variation of CTC concentration observed in our case was clearly due to infusion of trans-species cultured cells, and does not resemble the anticipated situation in patients. However, our observation does demonstrate that our system can identify differential CTC concentrations over time. Therefore, the methodology incorporated into our system, which permits patient-convenient CTC enrichment of a large volume of blood over a long time, and into which periodic sampling can be performed without multiple venipunctures, may provide insight into these biologically, and perhaps clinically, important questions.

Tumor biomarker assays may have clinical utility in one of several use contexts, including risk categorization, screening for undetected cancers at early stage, differential diagnosis, prognosis-independent of therapy, prediction of benefit of therapy, and serial monitoring to determine the state of the cancer[34,35]. Currently, CTC have been shown to have clinical validity as a prognostic factor in early stage breast cancer[36,37] and in the metastatic setting in several epithelial malignancies[38–41]. The reliability and higher recovery rate from long-term, large blood volume scans might broaden the clinical applicability of CTCs for many of the other use contexts, particularly through further downstream genotyping and molecular characterization[42–44] without the need for sophisticated CTC culture techniques[45].

Our first validation study of the aphaeretic CTC isolation system has produced promising results. However, several challenges remain before the methodology can be implemented into clinical settings. In theory, the system could potentially be exposed to several liters of blood in human circulation during the 2 h time of intravenous access. The CTC collection module, as presently designed, permitted interrogation of approximately 10–20 mL of blood in the dogs over 2 h, representing approximately 1–2% of the entire canine blood volume. Scaling up, this amount would translate to only 50–100 mL in an adult human. We are now working on improving the maximum flow rate through the system, conceivably by 3D stacking of the chips or by applying the system for a longer period of time with replaceable chips. In addition, the module used in our current system was designed to target the epithelial antigen for CTC immobilization, since we previously observed that many canine epithelial cancers express EpCAM[46]. However, since CTCs are highly heterogeneous[42,43,47], it is likely that we are missing subpopulations with reduced or loss of epithelial markers during epithelial-to-mesenchymal transition[48]. Applying a combination of various antibodies to coat the surface of the current capture module could potentially expand the capability to isolate a wider range of CTCs subtypes. In addition, since the system can fit any CTC isolation platforms with interchangeability, it will be of interest to test marker-agnostic devices[49] to elucidate whether CTC subgroups differ in their clinical implications in future studies.

In summary, we show that a temporary indwelling intravascular aphaeretic system can enable long-term operation in vivo to continuously harvest large quantities of CTCs, by re-transfusing the remaining blood products after the isolation procedure with minimal cell loss nor patient burden. Successful demonstration in canine models confirms the feasibility of our approach for future interventional clinical studies. The flexibility of the current system design can also be combined and adapted with various CTC enrichment methodologies or biochemical sensors that requires real-time analytical information from the blood. Finally, high numbers of CTCs obtained from large volume of blood screening will significantly reduce errors in determining the disease status and allow multiple characterization of CTCs to gain insight into their molecular and functional role realizing the full potential of a true liquid biopsy.

## Methods

**Antibodies**. Capture antibody: Anti-EpCAM antibody (BAF960, R&D Systems) 1/4 dilution (25ug/mL)

Primary antibodies: Anti-cytokeratin antibody (349205, BD Biosciences) 1/5 dilution, Anti-canine CD45 antibody (MCA1042GA, Bio-Rad) 1/50 dilution, Anti-canine CD45 antibody (MCA2035S, Bio-Rad) 1/10 dilution

Secondary antibodies: (A-21133,Invitrogen) 1/250 dilution, (A-21121, Invitrogen) 1/250 dilution, (A-11006,Invitrogen) 1/250 dilution

**Fabrication of system manifold and HBGO chip**. The overall schematics of the entire system are illustrated in Fig. 1. The design of the system manifold was created by CAD software (Solidworks) and fabricated using a high-resolution 3D printer (Projet 3500 Max). An acrylic based resin, M3 crystal, was used for the printing process for its mechanical integrity and biocompatibility. Each component including the pump, heparin injector, power source, and micro controller were placed in the designated compartment and enclosed (Fig. 1b)

The production of the HBGO chip involved two separate processes. First, to fabricate the chip substrate, Cr and Au were evaporated onto a 4 inch silicon dioxide wafer and patterned. The wafer was then diced into individual pieces. Next, to fabricate the PDMS structure, a silicon master mold was created by standard photolithography. Negative photoresist (SU-8 2050, MicroChem) was patterned on a 4 inch silicon wafer using two separate masks: one for the main fluidic channel (40 μm height) and the other for the herringbone grooves (60 μm). The height of each layer was measured after each process with a surface profilometer (Veeco Dektak 6 M). PDMS pre-polymer mixed with cross linker at a 10:1 weight ratio was poured onto the mold, degassed, and baked in an oven at 65 °C for 24 h. The cured PDMS structure was then carefully peeled off and cut. Finally, two through-holes were punched with a biopsy punch at both ends of the channel to feed and connect the tubing.

**HBGO chip assembly and surface functionalization**. To chemically modify the chip surface, tetrabutyl ammonium hydroxide intercalated graphene oxide nano sheets grafted with phospholipid-polyethylene glycol-amine were prepared and assembled on the gold patterned silicone dioxide substrate by electrostatic

attraction[27]. The substrates and PDMS replicas were subjected to oxygen plasma treatment and bonded to form the final device. N-γ-maleimidobutyryl-oxysuccinimide ester (GMBS) was flowed through the chip using a syringe pump (Harvard apparatus) and incubated for 30 min. The chip was then flushed with 70% ethanol to pre-sterilize the inner chamber wall. Subsequently, neutravidin and biotinylated anti-EpCAM antibody (BAF960, R&D Systems) was introduced followed by 3% bovine serum albumin (BSA) to block the remaining binding surface.

**Sterilization process.** To prevent microbial contamination, all disposable units within direct contact to the blood including the tubes, luer connectors, and syringes were sterilized using heat or ethylene oxide gas sterilization and packaged separately. For the CTC capture chip, the substrate was exposed to UV and the PDMS was autoclaved before assembly. All surface modification steps were performed in a germ poor environment. Each reagent was sterilized and tested for endotoxin level using limulus-amebocyte-lysate (LAL) gel clot assay (0.5 EU/mL sensitivity, Lonza) before use. After complete functionalization, the device was exposed to UV and the fluid within a subset of chips were sampled, plated on sheep blood agar, and cultured for 2 weeks to detect any bacterial growth.

**Cell culture and labeling.** The cultured human epithelial breast cancer cell lines, MCF7 (estrogen receptor positive, HER2 negative) and BT474, were purchased from the American Type Culture Collection (ATCC, LGC Standards) and were authenticated by the vendor. Both cell lines were tested for mycoplasma contamination using MycoAlert[TM] (Lonza) and Universal Mycoplasma Detection Kit (ATCC) and were found to be negative. MCF7 and BT474 cells were cultured at 37 °C with 5% $CO_2$ and maintained by regular passage in complete media consisting of Dulbecco's Modified Eagle's Media (DMEM) with 10% fetal bovine serum (FBS) and 1% penicillin-streptomycin solution (GIBCO®, Life Technology). When cells reached a confluency of 70–80%, they were collected and fluorescently labeled with CellTracker[TM] Green CMFDA dye (Invitrogen, C7025) or Cell-Mask[TM] Orange plasma membrane stain (Invitrogen, C10045) for cell capture experiments. For in vivo cell capture, MCF7 cells purchased from ATCC were transfected with NucLight Red[TM] (Essen BioScience, Ann Arbor, MI) according to manufacturer instructions (Supplementary Fig. 12).

**Cell viability assay.** To measure cell viability after processing samples through the [HB]GO chip, a live/dead viability/cytotoxicity assay kit (GIBCO®, Life Technology) was used. The chip was washed with 1× PBS (phosphate buffered saline, GIBCO®, Life Technology) after capturing the cells. Subsequently, a live/dead reagent consisting of calcein AM and ethidium homodimer-1 was prepared according to the manufacturer's instruction and applied. Following 30 min incubation, cells were imaged and manually counted under a fluorescent microscope for quantification.

**Human blood specimen collection.** Whole blood was drawn from healthy volunteers after obtaining informed consent for research testing that complied with the ethical regulations under an Institutional Review Board (IRB)-approved protocol at the University of Michigan. All samples were collected into CellSave tubes (Menarini Silicon BioSystems) and processed within 4 h.

**Canine model.** All canine experiments were performed with approval from the Colorado State University Institutional Animal Care and Use Committee (IACUC, 16–6490 A) for animal testing and research that complied with ethical regulations. Young adult male beagles (12–18-months-old) were purchased from a commercial vendor and housed individually in an IACUC-approved facility at the Colorado State University Veterinary Teaching Hospital with standard day/light, watering and feeding schedules. Baseline hematology, clinical chemistry, and coagulation profile were obtained. Animals were randomly chosen for each experiment.

For the ex vivo cell capture experiments, cultured MCF7 cells were heparinized, suspended in 5 mL of sterile pyrogen free NaCl and injected intravenously via the cephalic vein. Following injection, peripheral blood was serially collected via jugular venipuncture into CellSave tubes at multiple time points and evaluated for capture efficacy. A minimum of 3 independent experiments were carried out for all ex vivo studies.

For the in vivo cell capture experiments, the dog was sedated with butorphanol and acepromazine. The ventral cervical region was clipped and aseptically scrubbed followed by placing a 14-gauge dual lumen central venous catheter (Arrow/Teleflex, Morrisville, NC) in the jugular vein. The system was connected to the catheter with extension sets and primed. One minute after injection of MCF7 cells via the cephalic vein, the system was activated to allow blood to circulate through the CTC capture module for up to 120 min. Following completion, the central catheter was removed and the dogs were allowed to recover. The following day and 7 days later, blood was submitted for hematology, clinical chemistry, and coagulation profile. The dogs were not screened for the development of cancer since human MCF7 breast cancer cells are xenogeneic to dogs and not expected to colonize in an immunologically intact animal. Each dog was continuously monitored with daily examinations of temperature, pulse, respiration, food and water intake, and the catheter site for one week after the experiment. Then, the dog was off-protocol and eligible for adoption or transfer to other protocols as appropriate.

**Immunofluorescent staining.** After capturing on the [HB]GO chip, unlabeled MCF7 cells spiked into blood or injected into dogs were fixed with 4% paraformaldehyde (PFA). Before staining, the cells were permeabilized with 0.1% Triton-X followed by blocking with a blocking buffer containing 2% normal goat serum and 3% BSA. A cocktail of primary antibodies including anti-cytokeratin 7/8 (349205, BD Biosciences) and two types of anti-canine CD45 (MCA1042GA and MCA2035S, Bio-Rad) were diluted in 1% BSA and flowed through the chip. After a quick wash, secondary antibodies conjugated with Alexa Fluor 546 and 488 (A-21133, A-21121, and A-11006, Invitrogen) were prepared in 1% BSA and applied for probing. The cell nuclei were stained with 4',6-diamidino-2-phenylindole (DAPI, Invitrogen) as the final step before microscopic imaging. Cells were observed under ×10 magnification using the inverted epifluorescence microscope (Ti Eclipse, Nikon) with an automated motor stage. Images were reviewed blinded and analyzed manually (Image J and NIS-elements, Nikon).

**Reporting summary.** Further information on experimental design is available in the Nature Research Reporting Summary linked to this article.

## Data availability
The data that support the findings of this study are available from the corresponding author upon reasonable request. Source data for Figs 2–4, as well as the Supplementary Information figures are available in the Source Data file.

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

## Acknowledgements

The authors thank B. Meis and B. Rose for procurement of canine blood samples and cell line preparation and H. Kim for technical help with the development of the mobile application. This work was supported by the Susan G. Komen Foundation, the Fashion Footwear Charitable Foundation of New York/QVC Presents Shoes on Sale™, and the National Institutes of Health (NIH) Director's New Innovator Award (1DP2OD006672-01). The work was performed in part at the Lurie Nanofabrication Facility, a member of the National Nanotechnology Infrastructure Network, which is supported by the National Science Foundation.

## Author contributions

T.K., S.N., D.H.T., and D.F.H. conceived and designed the study. T.K. and C.R.O. built and tested the system with wireless controller. T.K., A.W. and K.J.S. fabricated the chips and performed the ex vivo CTC isolation experiments. T.K., Y.W. and D.H.T. performed the in vivo CTC capture experiments in canine models. D.H.T. provided canine blood samples and collected the patient data. L.C. and C.P. provided essential protocols and technical support. T.K., S.N., D.H.T. and D.F.H. analyzed the data and interpreted the results. T.K., S.N. and D.F.H. co-wrote the manuscript. All authors discussed and edited the manuscript.
