## [Peer Review File · Nature Communications]

Reviewers' Comments:

Reviewer #1:

Remarks to the Author:

This manuscript describes an important advance in the in vivo analysis of circulating tumor cells, where an indwelling device is used to collect and measure CTC levels. The CTC enumeration measurements have impressive precision, and the authors make the case that this one of the justifications for the method. The following comments address points that could be strengthened.

- 1) Given that what is reported is an engineering-focused integration effort, the work would be significantly strengthened if the authors showcased an application that cannot be realized with in vitro analysis, or elucidated interesting new biology with the approach.
- 2) The authors indicated that they injected 20,000,000 MCF7 cells, and recovered ~ 700. Is this cell type specific? It would be helpful to know if the MCF7 cells they are use are a particularly robust model.
- 3) The workflow that is used for the measurements is not clearly laid out in the manuscript. This should be sketched out in Figure 1.
- 4) How do the authors envision taking into account the heterogeneity of CTCs? The approach appears to suffer from many of the limitations of other immunoaffinity methods in this regard.
- 5) Are CTC clusters detectable using the approach? Again, more validation using cultured cells may help elucidate this.

Reviewer #2:

Remarks to the Author:

Circulating tumor cells (CTCs) are rare events which poses a limitation to their further downstream analysis. Over the past 5 years, the first in vivo detection and capture systems have been developed with the promise to increase the yield of CTC capture and monitor the circulating tumor burden in cancer patients. This is a very important area of translational cancer research. Here, the authors have developed a new tool for the in vivo enrichment of circulating tumor cells (CTCs) and tested the function of this device in beagle dogs. The device is able to analyze 1-2% of the total blood and - based on spiking experiments with the MCF7 breast cancer cell line – it results in higher capture rates than sequential blood draws performed in parallel. The study is well performed by an experienced team of researchers but it still has some limitations.

Specific comments

- (1) In vivo capture of CTCs has been performed by the nanowire from a company called GILUPI (Germany) and the authors have cited the key publications (e.g., Gorges et al., CCR 2016). The authors claim a higher yield of their new device compared to other in vivo devices but no direct comparison was performed. Compared to their serial blood draws, the capture rate of their in vivo device was 3-4 times higher; this value is not so different from reports on other in vivo devices or reports on new in vitro devices or new CTC markers.
- (2) The device can screen 1-2% of total blood. Although no data on humans were provided, this would be an estimate of approx. 50-100 mL blood, an amount that could be taken from cancer patients for diagnostic purposes. It would be important to further increase the blood volume.
- (3) The authors used a dog model and report no side effects. However, I could not find the animal welfare allowance and the information how long the dogs were screened for the development of cancer.
- (4) The device is certainly very interesting but the use in cancer patients might reveal problems that are not envisaged by the current analysis of spiking experiments performed with one breast

cancer cell lines. The heterogeneity of cancer cells from patients in clinical samples makes their identification much more challenging than spiking experiments usually indicate. MCF7 cells are larger and have a strong keratin expression and are therefore easily detectable in blood specimens. Other in vivo devices have been used in cancer patients (e.g., GILUPI device) and it might be difficult to claim superiority based on the current experiments in dogs.

(5) I could not find any information on the capture rate/efficiency of the in vivo device, i.e., how many breast cancer cells were injected and how many were captured?

(6) Serial sampling was performed using small volumes of 1mL, which might explain the lower detection rate.

(7) The work lacks a comparison with an established CTC technology (e.g., the CellSearch technology co-developed by the senior authors).

Reviewers' comments:

Reviewer comments are *italicized* while our responses are in **red text**.

Reviewer #1 (Remarks to the Author):

This manuscript describes an important advance in the in vivo analysis of circulating tumor cells, where an indwelling device is used to collect and measure CTC levels. The CTC enumeration measurements have impressive precision, and the authors make the case that this one of the justifications for the method. The following comments address points that could be strengthened.

We appreciate the Reviewer's support for our manuscript. Based on the comments, we incorporated several edits in the revised manuscript as detailed below.

1) *Given that what is reported is an engineering-focused integration effort, the work would be significantly strengthened if the authors showcased an application that cannot be realized with in vitro analysis, or elucidated interesting new biology with the approach.*

We thank the Reviewer for the suggestions to strengthen our work. As noted by the Reviewer, our hypothesis is that *in vivo* isolation of CTCs may provide many opportunities not available with standard *ex vivo* blood processing and analysis, since it permits interrogation of larger volumes of blood, and therefore in theory allows analysis of larger number of CTCs, which are released from their primary tissue source over a long period of time. Thus, our Proof-of-Principle results open the possibility to more accurately and easily perform several applications, such as enumeration of CTC with higher statistical confidence, and perhaps more importantly molecular phenotyping, genotyping, and even culturing CTCs. Each of these has been reported by us, and others, with available *ex vivo* assays, but the success in doing so has been modest, at best, due to limited CTC samples.

We have not conducted any of the molecular analyses, since we used known MCF7 cell lines in our artificial canine model to establish the validity of the system for *in vivo* isolation. We suggest that further downstream analyses of those cells, which have been well characterized over decades, may not be meaningful in the context of our study. Although we can do so, the availability and number of investigational animals we are permitted to include in each

experiment is limited. However, in the revised manuscript (page 13), we have enhanced our discussion of potential applications of the technology for future clinical use as shown below. A more extended study using canine patients with spontaneous cancers is ongoing and we anticipate showing new biological insights or specific applications in our follow-up studies.

“Tumor biomarker assays may have clinical utility in one of several use contexts, including risk categorization, screening for undetected cancers at early stage, differential diagnosis, prognosis independent of therapy, substitution or prediction of benefit of therapy, and serial monitoring to determine the state of the cancer.”

2) *The authors indicated that they injected 20,000,000 MCF7 cells, and recovered ~ 700. Is this cell type specific? It would be helpful to know if the MCF7 cells they use are a particularly robust model.*

MCF7 cells were chosen to mimic the presence of CTCs in our animal model due to their known epithelial cell adhesion molecule (EpCAM) expression which is commonly observed on CTCs of epithelial origin, but not on blood cells. To demonstrate *in vivo* capture of these cells after injecting in canine, the CTC capture module was specifically coated with antibodies against EpCAM. Thus, our model is sufficiently robust to demonstrate that this first-generation *in vivo* system does, indeed, capture CTC from a living animal. As the Reviewer pointed out, this cell line may not represent or cover the broad spectrum of all cancer cell types. However, depending on the cell type of interest, the capture module can be freely modified using different targeting strategies which is one of the strength of our system design.

As anticipated, the capture efficiency in this xenograft model using an investigational animal with a fully intact immune system is quite low, since the vast majority of human MCF-7 breast cancer cells are cleared rapidly by the dogs' reticuloendothelial system. The CTC capture curves reflect this rapid clearance, but do, nonetheless, demonstrate the capability of capturing circulating epithelial cancer cells from whole blood *in vivo*. Importantly, as shown by our data, the *in vivo* method using our system yielded higher numbers of CTCs captured compared to the *in vitro* sampling approach.

No changes made to manuscript

3) The workflow that is used for the measurements is not clearly laid out in the manuscript. This should be sketched out in Figure 1.

We appreciate the opportunity to clarify the workflow of our experiments. Since the measurements were performed mainly in the animal model, we included a detailed schematic illustrating the workflow of the *in vivo* CTC capture in Figure 4a. (as shown below) in our revised manuscript.

Figure 4. Quantitation and comparison of MCF7 cells detected by ex vivo versus *in vivo* CTC isolation method.

4) How do the authors envision taking into account the heterogeneity of CTCs? The approach appears to suffer from many of the limitations of other immunoaffinity methods in this regard.

We agree with the Reviewer that not all CTCs may express EpCAM and thus the current form of our CTC capture module may not be able to capture all type specific cancer cells in a patient's blood. The purpose of using the anti-EpCAM based strategy was to provide demonstration of isolating MCF7 cells in our animal model. However, the chip portion of the system can simply be modified using a number of possible strategies, such as functionalizing the surface with combinations of various antibodies to target a diversity of potential surface markers other than EpCAM. In addition, since the CTC capture module is freely interchangeable, chips based on

filtration or flow dynamic differences between cancer and normal hematopoietic cells can be adapted. In these regards, our modular design approach has the potential to capture an expanded variety of circulating biomarkers taking into account the heterogeneity of CTCs. We revised our manuscript to acknowledge these points in page 14 as below.

“Applying a combination of various antibodies to coat the surface of the current capture module could potentially expand the capability to isolate a wider range of CTCs subtypes. In addition, since the system can fit any CTC isolation platforms with interchangeability, it will be of interest to test marker agnostic devices subsequently to elucidate whether CTC subgroups differ in their clinical implications in future studies.”

5) Are CTC clusters detectable using the approach? Again, more validation using cultured cells may help elucidate this.

The biological and potential clinical significance of CTC clusters is indeed of interest in the field. However, since the amount of artificial CTC clusters formed using cultured cell aggregates are difficult to control and the clustering mechanics differs from that observed in patient's blood, enumeration and quantifying the capture rate was not within the scope of our report. Of note, we have tested whether the shear forces experienced by artificial MCF7 cell aggregates exceeds the capacity to maintain cell viability in our *ex vivo* experiments. If the field evolves to demonstrate that the presence of clusters provides clinical utility, we are confident that our device can be optimized to specifically capture and analyze CTC clusters.

No changes made to manuscript

Reviewer #2 (Remarks to the Author):

Circulating tumor cells (CTCs) are rare events which poses a limitation to their further downstream analysis. Over the past 5 years, the first in vivo detection and capture systems have been developed with the promise to increase the yield of CTC capture and monitor the circulating tumor burden in cancer patients. This is a very important area of translational cancer research.

Here, the authors have developed a new tool for the in vivo enrichment of circulating tumor cells (CTCs) and tested the function of this device in beagle dogs. The device is able to analyze 1-2% of the total blood and - based on spiking experiments with the MCF7 breast cancer cell line – it results in higher capture rates than sequential blood draws performed in parallel. The study is well performed by an experienced team of researchers but it still has some limitations.

We thank the Reviewer for the useful comments which improved the quality of the manuscript. In light of these remarks, we conducted additional experiments and made several changes to address the points listed below.

Specific comments

1) In vivo capture of CTCs has been performed by the nanowire from a company called GILUPI (Germany) and the authors have cited the key publications (e.g., Gorges et al., CCR 2016). The authors claim a higher yield of their new device compared to other in vivo devices but no direct comparison was performed. Compared to their serial blood draws, the capture rate of their in vivo device was 3-4 times higher; this value is not so different from reports on other in vivo devices or reports on new in vitro devices or new CTC markers.

We again emphasize the Proof-of-Concept nature of our report. The primary objective of this work was to establish a proper CTC animal model and demonstrate successful capture of human breast cancer cells that have been injected in canine at a distant intravenous site using a miniaturized in vivo CTC isolation system. Thus, we have focused on performing studies whether the system can be safely placed and left intact for several hours in animals without adverse effects and have limited our comparison of CTC capture yield to ex vivo analysis of sampled blood specimens.

Unlike *ex vivo* CTC devices, it is difficult to directly compare the yield due to the differences in testing conditions including time, subject model, vascular access site and standardization in measurements. A recent study has shown that the GILUPI CellCollector is only capable of capturing the non-small cell lung cancer cell line H1650 with an overall efficiency of 0.0016 ± 0.0003 % using an *ex vivo* blood circuit model^[1]. However, this yield underestimates the approach as it does not simply translate to the device capability of isolating CTCs in patients which is also true with our system.

Compared to the few technologies that have been previously published, our system has the advantage of long term and swappable interrogation ability. In addition, the capture rate of our system does not depend on the blood access site, while the GILUPI method highly depends on the vascular dimensions since it passively captures CTCs by contact, which is an added advantage.

We did not intend to imply that our system is clearly superior to existing *in vivo* devices. Thus, we have revised the manuscript to only reflect that our system results in a higher CTC capture yield than using the same ^{HB}GO chip *ex vivo*, based on EpCAM expression capture.

“The total capture of CTC in vivo by the indwelling system was approximately 3.5 times that compared to the periodic blood draw approach.” (page 10)

In vivo isolation of CTCs is a nascent field and we believe that our technology integrating the engineering components to enable monitoring CTCs using a miniaturized wearable device is innovative. We anticipate that with further optimization, our system will have increased flow, thus increasing the volume of blood to be interrogated, without sacrificing capture efficiency.

Nonetheless, to be comprehensive, we have included a recently reported *in vivo* CTC isolation technology similar to the GILUPI CellCollector but which utilizes magnetic particle injection for pre-labeling CTCs and applying a wire with magnetic force to actively capture cells^[1] in our reference (ref #26) of the revised manuscript. Therefore, we have added the following sentences to our Introduction, page 5:

“Similarly, a recent study has demonstrated in vivo capture of non-small cell lung cancer cells injected into a porcine model, using a flexible magnetic wire (MagWIRE)²⁶. However, the

approach requires pre-injection of EpCAM coated magnetic particles to label CTCs which limits its long-term application due to possible systemic exposure of iron overload.”

2) The device can screen 1-2% of total blood. Although no data on humans were provided, this would be an estimate of approx. 50-100 mL blood, an amount that could be taken from cancer patients for diagnostic purposes. It would be important to further increase the blood volume.

We agree with the Reviewer that the volume of blood interrogated may not be optimal with the current design. To translate the technology to clinic, we would like to sample much larger volumes of the blood. However, the scope of this work was focused on developing the concept and validating its use for *in vivo* CTC isolation in a xenograft animal model. Ongoing optimization studies are designed to increase the flow/unit time over the capture module so that larger volumes of blood are interrogated as noted above. However, the current data, although limited by throughput, clearly demonstrates the feasibility of this approach.

Further, as noted in the original manuscript's Discussion section, we observed that not only the total blood volume may be of interest since CTC shedding can occur at different time periods, although further verification is necessary. Thus, the capability to analyze blood for a prolonged period of time may influence the capture rate of CTCs and the work we propose may have added advantages over single blood sampling of more than 50-100 mL.

No change made to manuscript.

3) The authors used a dog model and report no side effects. However, I could not find the animal welfare allowance and the information how long the dogs were screened for the development of cancer.

We thank the Reviewer's attention to details. The CSU IACUC number is 16-6490A and each dog was continuously monitored with daily examinations of temperature, pulse, respiration, food and water intake, and the catheter site for one week after the experiment. The dogs were not screened for the development of cancer since MCF7 cells are xenogeneic to dogs and have no greater chance of growing than transplanting a human organ. This was one of the reasons for choosing this cell line for our animal model. We have included this information in the updated version of our manuscript (page 18-19 of the method section).

“All canine experiments were performed with approval from the Colorado State University Institutional Animal Care and Use Committee (IACUC, 16-6490A).”, “The dogs were not screened for the development of cancer since human MCF7 breast cancer cells are xenogeneic to dogs and not expected to colonize in an immunologically intact animal.”

4) The device is certainly very interesting but the use in cancer patients might reveal problems that are not envisaged by the current analysis of spiking experiments performed with one breast cancer cell lines. The heterogeneity of cancer cells from patients in clinical samples makes their identification much more challenging than spiking experiments usually indicate. MCF7 cells are larger and have a strong keratin expression and are therefore easily detectable in blood specimens. Other in vivo devices have been used in cancer patients (e.g., GILUPI device) and it might be difficult to claim superiority based on the current experiments in dogs.

We agree with the Reviewer's concerns, which are similar to those of Reviewer 1. We chose MCF7 cells specifically because they are known to highly express EpCAM, which is the target of our antibody-based capture system for these experiments. Of note, the rationale behind using the anti-EpCAM base method in our system stem from our preliminary study observing that most canine carcinomas have shown high EpCAM expression^[2]. However, we emphasize that the capture strategy can be altered using various methods as the CTC capture module is fully interchangeable.

In our previous studies, we have demonstrated that cells with both high/low EpCAM expression can be efficiently be capture using functional graphene oxide^[3], which has also been used in our ^HBGO chip module. In addition, since the device incorporates a channel design that is capable of directing the cells toward the antibody coated substrate, cells of different size should not significantly affect the capture performance of the system.

As mentioned in the response to Question #4 of Review 1 above, we are aware that CTC heterogeneity is a problem with any capture chip. However, again, we propose that now that we have established Proof-of-Principle with the current system, these issues can be addressed with further optimization. The flexibility of our modular system design will allow ease of swapping the capture modules for these following investigations. We are currently in the process of conducting such experiments in pet animals with spontaneous cancers to investigate different

targeting strategies that may result in an improved CTC capture rate which cannot be done using cultured cell lines.

Changes in revised manuscript as per response to Reviewer 1, Comment 4 above.

5) I could not find any information on the capture rate/efficiency of the in vivo device, i.e., how many breast cancer cells were injected and how many were captured?

As the Reviewer points out, the capture efficiency for an *ex vivo* device that interrogates a fixed, and relatively low volume of blood is important. However, we argue that capture efficiency for an indwelling, intravascular, *in vivo* device is less critical, since our device will interrogate relatively larger volumes of blood over longer periods of time.

Indeed, we agree with the Reviewer that higher capture efficiency will increase the number of cells, and now that we have established successful capture and enumeration with the current prototype, we are working on the next generation system that will accommodate larger blood volumes. Nonetheless, we provide the following calculation in response to the Reviewer's query:

The total number of MCF7 cells injected in canine was 2×10^7 cells and the total number of cells captured *in vivo* was 762 cells/2 hours (=0.0004%). However, the xenograft animal model does not permit an accurate capture efficiency calculation, as the cell concentration in circulation varied as a function of time (the vast majority of human MCF-7 cells were removed within a few hours by the immunologically intact animal's reticuloendothelial system).

To clarify the Reviewer's point, we conducted an *ex vivo* experiment to calculate the relative efficiency of CTC capture using the system and included the following details in the revised Supplementary Information. Prelabeled MCF7 cells were spiked into buffer with a concentration of 200 cells/mL and pumped through a tube (diameter of 4 mm) at a flow velocity of 1 cm/s using a peristaltic pump to mimic the blood flow with a steady-state CTC concentration. The system was connected to a catheter which was introduced into one side of the tubing and operated for 30 mins. The total number of cells captured on the ^{HB}GO chip was 576 cells (**Figure S9a**). This was similar to the number of MCF7 cells captured by processing the same volume of solution through a ^{HB}GO chip *ex vivo* (543 cells, **Figure S9b**), indicating that the amount of CTC capture is simply proportional to the blood processing volume from the system.

This is one of the advantages of our system since the capture efficiency does not depend on the physiological conditions (vein diameter or blood flow rate) of the human or animal subject. Therefore, it is easier to standardize the CTC counts in blood compared to other *in vivo* approaches which highly depends on the vein of access. Overall, the number of CTCs detected through this 30 min operation resulted in a capture efficiency of approximately 1 %. However, we emphasize that these data are relatively irrelevant to the actual performance of an indwelling CTC capture device for the reasons mentioned in the response to the Reviewers' questions #1.

Figure S9. Experimental setup to quantify the CTC capture efficiency. a. The system was connected to a tube, mimicking the blood flow in a vein, using a dual lumen catheter. b. The same cell containing solution was prepared and injected into the ^{HB}GO chip for comparison. c. Microscopic scanning images of the chips used in the two experiments.

In addition, since the system's capture rate was proportional to the efficiency of the CTC capture module, we conducted an additional experiment using MCF7 cells spiked into canine blood which may provide a better understanding of the overall performance. The capture rate of the chip module was $83.23 \pm 5.89 \%$ and $62.97 \pm 7.83\%$ at a flow rate of 50 and 100 $\mu\text{L}/\text{min}$ respectively (**Figure S6.**). We have included these data in the revised supplementary information, page 10.

Figure S6. Capture efficiency of ^{HB}GO chip for MCF7 cells spiked in canine blood and processed *ex vivo*.

6) Serial sampling was performed using small volumes of 1mL, which might explain the lower detection rate.

As the Reviewer mentioned, the small volume of serial sampling used for *ex vivo* CTC capture compared to our *in vivo* method may have caused a lower detection rate. However, this exactly demonstrates why it is necessary to interrogate larger blood volume for CTC analysis, which not only increases the CTC detection rate but also provide a high statistical confidence in quantification. The 1 mL of sampling volume was roughly the amount of blood used for classical *ex vivo* CTC isolation technologies (such as CellSearch) that has been scaled to the total blood volume of a dog.

No changes made to manuscript.

7) The work lacks a comparison with an established CTC technology (e.g., the CellSearch technology co-developed by the senior authors).

In the long run, the Reviewer is absolutely correct. However, the purpose of our work resides on demonstrating the capability to interrogate larger blood volumes to capture more CTCs than a single blood draw, which we alternatively have shown using the same capture chip *ex vivo* for comparison, as the ^{HB}GO chip already allowed high sensitivity. Also, since all *ex vivo* methods, such as CellSearch, are fundamentally limited by the finite blood sampling volume, we suggest that this would be another *in vivo* to *ex vivo* comparison as we extend the experiments to longer

time frames. However, once we have optimized the performance of the flow through the system and chip module (which is the focus of our next round of experiments), future investigations will be designed to compare results of our system with previously existing systems in a single canine with spontaneous cancer.

No change made to manuscript.

Reference

- [1] Vermesh, O. *et al.* An intravascular magnetic wire for the high-throughput retrieval of circulating tumour cells in vivo. *Nature Biomedical Engineering* 1–13 (2018).
doi:10.1038/s41551-018-0257-3
- [2] Thamm, D. H., Hayes, D. F., Meuten, T., Laver, T. & Thomas, D. G. *Epithelial Cell Adhesion Molecule Expression in Canine Tumours.* *J. Comp. Pathol.* 155, 299–304 (2016).
- [3] Yoon, H. J. *et al.* Sensitive capture of circulating tumour cells by functionalized graphene oxide nanosheets. *Nat Nanotechnol* **8**, 735–741 (2013).

Reviewers' Comments:

Reviewer #1:

Remarks to the Author:

The revised manuscript was improved in response to the suggestions of the reviewers. Additional data was included on CTC clusters that enriched the conclusions that could be drawn. A key point - on demonstrating how this technology uniquely satisfies an unmet need - was not addressed, but the overall value proposition of what is described may overcome this objection.

Reviewer #2:

Remarks to the Author:

The authors have addressed my comments and improved the manuscript. I have no further comments.